# The Trailer as Erotic Capital. Gendered Performances—Research and Participant Roles during Festival Fieldwork

**Irene Trysnes** 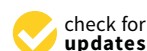

Department of Sociology and Social Work, University of Agder, 4604 Kristiansand, Norway; irene.trysnes@uia.no

**Abstract:** This article examines different roles and field relations of the researcher in studies of young people at Christian festivals. The main questions are how the researcher gains access to the "flirtation field," which flirting roles the youth participants engage in, and how the researcher copes with flirtation in the field. The article's theoretical approach draws on feminist methodology on how positions, roles, and relations are negotiated in fieldwork, and discusses the notion of erotic capital. Christian festivals in Norway attract among 100,000 people every year. One of the main activities that was brought to my attention by a car trailer was the importance of flirting between the young boys and girls at these festivals. At the Christian festivals, flirting takes place within a heterosexual framework. In order to be part of this game, the girls are supposed to be feminine and available. The boy's role is to be active and take initiative. Both sexes work hard to become participants in this game of winning attention that represents two different worlds for boys and girls, and in which there are both male and female losers. The rules of the game seem doubly strict for the girls since they are expected to administer both their own and the boys' lust.

**Keywords:** flirting 1; gender 2; festivals 3; youth 4

## 1. Introduction

It's almost 2 a.m. I'm on my way to my tent to go to sleep when I'm stopped by a group of boys along the road.

"Hey, you! You with the striped sweater," they shout.

"Yes?" I turn around and nod at them. They look at me. I feel judged by their looks. Then they look at each other, smile, and nod.

"Would you like to come over and stretch out with us?"

"Excuse me?" I say. They laugh.

"Come over and stretch out with us," another boy says. One of them does something resembling the splits. They laugh again. "Come on then!"

"I see you're very good at stretching," I say. "Thanks for the kind offer, but I'm going to bed."

"Are you sure?"

"Absolutely sure!" I say. "But thanks for asking." I walk quietly back to my tent while I hear the boys yelling after me and laughing.

(Field note from Christian festival, my translation).

The above episode and other similar incidents during my fieldwork at Christian festivals made me aware that a researcher is not just a researcher. In this particular fieldwork, gender and age became of crucial importance to gain access to, for instance, the flirting arena.

Christian festivals in Norway attract a considerable number of people. Each year around 100,000 people gather at these festivals, and most religious organizations host their own events. The empirical

material is conducted from fieldwork from seven Christian youth festivals situated in the southern and western part of Norway. The festivals represent different activities, themes, and also theological views. They therefore also attract different types of participants. Some consider themselves to be very religious, while others are not interested in religion at all.

One matter of great concern when young people come together at huge festivals is flirting and the idea of romantic love [1]. Christian festivals differ from other festivals because they focus on religious experiences and they also have strict guidelines on sexual behavior. To prevent sexual activities, most Christian festivals do not allow boys and girls to sleep in the same tents. Some festivals have separate seminars for boys and girls where they educate them about sexual behavior and boundaries. The framework of marriage between man and woman is also a way of presenting and discussing sexuality, and the concept "worth waiting for" is crucial [2] (p. 167). How does the researcher gain access to the "flirtation field" and how does one cope with flirtation in the field? This article examines different research and participant roles and gendered performances in studies of young people at Christian festivals. Studying flirting strategies in Christian festivals is of sociological interest because this is a field that represents different kinds of rules and social norms than what many of the young people meet in other parts of society.

Festivals are often related to celebration, experiences, and leisure [3]. The festival is limited to a certain time, which makes it more intense and could even be said to represent a "universe on its own." Further, I will argue like Kaspar and Landholt [4] that there is a need for more knowledge on how the researcher is affected by sexual attention in fieldwork:

"[E]nactment of sexuality in fieldwork is far more common than is reflected in the current body of literature, and then even apparently innocuous sexualizations have a considerable effect on the way gender and sexuality are negotiated during research encounters, and thus the collection of data" [5] (p. 108).

Flirting can be described as a kind of interaction of excitement where the participants do not know what will happen in the next moment, and it represents a kind of risk behavior [6]. Flirting is also a gendered game, with different rules for men and women, and represents an "embodied social interaction that adds a sexual component to a mostly one-on-one interaction, in which sexual attraction to the interlocutor is expressed" [4] (p. 108). Studies of flirtations also indicate that it is often men who take the first step [1,7].

A microsociological approach on the study of flirting as a game of erotic capital and hegemonic masculinity.

This article's theoretical approach draws on feminist methodology on how positions, roles, and relations are negotiated in fieldwork and how the production of knowledge is situated and embodied, and interprets flirting as gendered performances [4,5,8]. Flirting is part of a social situation with certain expectations concerning being either a male or female. Sociologist Erving Goffman is known for using a wealth of different metaphors to describe social life, and some of them are well suited to analyzing flirting as a game. Goffman compares social interaction to a theatre in which individuals play roles and appear as performers in a ritual game. The actors can commit to a role, embrace a role, try out boundaries for the role, give it new content, make fun of it, isolate themselves from it, go into it as a joke, demonstrate that they have mastered it, and distance themselves from it: "The image that appears of the individual is a juggler and synthesizer, a mediator and reconciler who fulfills one function while he seems to be doing another" [4] (p. 212). Goffman [9] further divides the role performances between the *backstage* and the *front stage* [9] (pp. 107–112). The front-stage action is a type of façade performance where one shows off one's best side, and can be viewed as a public or professional performance.

Gendered codes must be learned in order to participate in the flirting game. With Goffman [6], gender functions as a framework that helps to structure and set guidelines for human interaction. Because gender is linked to the framework concept, it is also about power relations. Gender structures are thus at the core, and they govern human behavior concerning how appropriate it is to act in a

given situation. The framework for how women and men should act differs. Through the socialization process, various feminine and masculine portraits are constructed. Goffman has been criticized by recent gender theorists both for his role concept and for his lack of interest in studies of how gender structures affect and limit the performer's leeway [10,11]. Gardner [12] (p. 56) suggests that, when it comes to a theory of gender:

"there are two Goffmans, the first who wrote somewhat blithely, if not slightingly, of women's concerns in his books on behavior in public (1963 and 1971), and the second who built a superstructure of gender concerns designed to express women's differential treatment, most notably in his work on gender in interaction (Goffman 1977, 1979)."

Theorists like Judith Butler mainly have criticized the first Goffman and further contributed to develop the performative construction of gender [10].

To elaborate on the notion of erotic capital, one way of understanding it is to look at attractiveness. People who are considered to be attractive are also judged more positively than people who are considered unattractive, and people also seem to agree upon who is and who is not attractive across cultures [5,13]. However, Hakim [8] argues that erotic capital is a wider notion that "( . . . ) is just as important as economic, cultural, and social capital for understanding social and economic processes, social interaction, and social mobility" (p. 499). Erotic capital is also gendered: "Women have more erotic capital than men in most societies because they work harder at personal presentation and the performance of gender and sexuality" (p. 504). Gender is also the reason why erotic capital has been overlooked because the focus in earlier research has mainly been "on male activities, values and interests" [5] (p. 510).

When looking at dating and flirting in Christian festivals, it is crucial to understand how it proceeds and which processes include or exclude the young people from this game. Here, I argue that an understanding of erotic capital will be fruitful as an approach, together with theories of flirting as a gendered game [6]. Hakim suggests that erotic capital consists of seven different elements, all dependent on the cultural context. These elements are beauty, sexual attractiveness, social skills, liveliness, social presentation, sexual competence, and also, in many cases, fertility [8] (pp. 500–502). These elements are quite different from each other; for instance, how one views beauty might be in great contrast to how one views sexual attractiveness. When it comes to social skills, Hakim refers to the work of Arlie Hochschild. Hochschild has developed the term "feeling rules" and "emotional labor" to describe how institutions govern and regulate emotions [14] (p. 56). Hochschild [15] argues that women have more training in performing emotional labor than men. At the same time, women have stricter "acting manuals." This is because "( . . . ) subordinates having less control over their lives, must orient themselves to those in control" [11] (p. 291). When it comes to erotic capital, Hakim's point is that these elements play together and create a person's erotic capital in different ways. Erotic capital is something a person might both develop and perform in social interactions.

When studying young people in flirting relations, it is important to reflect upon how those relationships occur, and also how, as a researcher, one can gain access to the field in order to be able to study such relationships. Hakim's concept of erotic capital arises, as I see it, partly as a critique of sociological theories focusing only on class, gendered power relations, and the subordination of women. Erotic capital is a way of describing how these gendered relationships can be more complex, but also empowering for a female position. However, Hakim [8] seems little interested in whether the possession of erotic capital in itself can pin both men and women into certain roles or positions. The argument of women working harder at their gender performance and sexuality also indicates that erotic capital does not occur in a powerless vacuum. This further leads to the question of when flirting, or the erotic capital turns into being considered sexual harassment? Sexual harassment, according to Meyer [16] (p. 109), represents different types of gendered harassment.

Despite this critique of Hakim, I find the concept useful, but it needs to be combined with a theoretical understanding of gendered power relations. The concept of hegemonic masculinity conceptualized by Connell [17,18] is a fruitful approach in this context. The term "hegemonic

masculinity" is used to describe the dominant form of masculinity in Western society. Hegemonic masculinity "( … ) is always constructed in relation to various subordinated masculinities as well as in relation to women" [17] (p. 183). Connell argues that there exists no equivalent female hegemonic femininity, but one form is centered on the subordination of men "( … ) and is oriented to accommodating the desires of men" [17] (p. 183). Connell calls it "emphasized femininity" [17] (p. 183). However, hegemonic masculinity is not "( … ) a fixed character type, always and everywhere the same. It is, rather, the masculinity that occupies the hegemonic position in a given pattern of gender relations, a position always contestable" [18] (p. 76). This means that the hegemony is contestable.

I argue that the notion of erotic capital combined with Connell's understanding of hegemonic masculinity is able to capture both the power relation between young boys and girls, and flirting as a performance between young boys and girls at Christian festivals.

## 2. Materials and Methods

I attended the festivals as a participant observer. Three of the festivals were attended twice. I visited the others just once, due to time constraints. At the festivals, I stayed in a tent or cabin and tried to observe most of the activities going on. In the study of festivals I have used a mixed method approach " … for the broad purposes of breadth and depth of understanding and corroboration" [19] (p. 123). I have used a combination of field observation and qualitative interviews. In addition to this, I have also collected written material in the form of brochures, programs, and other archival material.

During my fieldwork, I communicated with about 300 different participants. In every festival, I tried to contact different groups of youth in different settings. Some youths attend all the religious activities, others do not consider themselves to be religious and do not attend the Christian meetings but just socialize with their friends.

In addition, I also talked to leaders during meetings and festivals. After the festivals, I conducted 36 semi-structured interviews with 36 leaders and participants. I used the snowball sampling method to recruit participants and leaders for interview. The youth participants and most of the youth leaders were recruited during the festivals. I also interviewed some leaders before the festivals started. In the recruitment process I tried to get in touch with different types of participants with different connections to the Christian festivals. Interviews and field notes were transcribed and anonymized. I do not mention the roles and duties of the informants in combination with names of the festivals in order to ensure anonymity.

The profiles of the festivals vary. Arena was the first festival I attended. This is a Christian sports festival, arranged by KRIK—Kristen idrettskontakt (Christian Sports Contact) and has an ecumenical profile. Second, I visited The Scandinavian Summer Camp and, third, a festival called Jesus to the Nations, both of which are arranged by the missionary organization Troens Bevis, a Pentecostal charismatic organization. The third is called Teen Mission Festival, arranged by The Norwegian Mission Society. The fifth is called Skjærgårds Music and Mission Festival. This is arranged by The North Mission, the children and youth organization of the Evangelical Lutheran Free Church, Changemaker, a youth organization connected to the Norwegian Church Aid, and Agder Episcopal Council. I also visited an art and dance festival organized by Youth With A Mission (YWAM) and, finally, a music festival called Island Gospel, an ecumenical festival held on the island Flekkerøya in the southern part of Norway. These festivals have approximately 14,000 participants. Most of them are young people aged 15–25. Due to the anonymity of the participants and the leaders I have talked to, I chose not to connect any of the field notes and quotes to the different festivals.

The main focus of the study was to get a broad understanding of what was going on at these festivals, inspired by the microsociological work of Erving Goffman. One of the main activities that was soon brought to my attention was the importance of flirting between the young boys and girls. The next part of this article will focus on how to observe flirting from a researcher perspective and how, even as a researcher, one is being flirted with in different fieldwork situations.

## 3. Results

### 3.1. It Started with a Trailer

The field worker's interaction in the field—who she meets and what areas she gains access to—will affect what she sees and her interpretations of what is going on. During fieldwork, the researcher will be assigned different types of roles and the roles will decide where the field worker can go, what she can do, who to cooperate with, what she can ask, what she can see, and what information she can be provided with [20] (p. 29). At the Christian festivals, it was important for me to acquire roles that also gave me the ability to observe backstage behavior [9].

"The fieldworker is always a marginal person, an outsider who, if he is successful, is permitted relatively free access to the backstage area of the local scene" [21] (p. 248).

I realized early on that being relatively young gave me an advantage. I was able to blend in with at least some of the oldest of the participants. In addition, being female in this context seemed to be an asset. Many of the male leaders and participants were very helpful in showing me around, and they also invited me to different types of arrangements within the festival. I was also offered all manner of practical assistance. Someone offered to help me park my car (implying that women cannot) and to put up the tent for me (implying that women cannot do practical work). In these cases, I politely rejected their offers, but it made me aware of how being a female researcher opened some doors and closed others.

( . . . ) small, female researchers are likely to appear to the subjects of their research as in need of assistance and instruction. They thus have different experience of fieldwork from that of the older, male, professorial "expert" who does not appear to be in need of being taught [22] (p. 65).

All the festivals attract many young people. After some time, I observed that many of the participants were not attending any of the Christian meetings or activities. I was interested in finding out what these groups of youths were doing all day, and I spent some time outside the meeting area. One important activity for these participants was, as they put it themselves, flirting. How does the researcher gain access to the flirting arena?

At one of the festivals, I coincidentally discovered something that opened my eyes to the flirting arena. I arrived at the festival in my car. Since I was going to stay for some time, I had a lot of luggage and therefore brought a small trailer. The first thing to happen was that I was pulled aside by a security guide and a police officer at the entrance point of the festival. I asked if I had done something wrong, but they just laughed without answering my question. Instead, the police officer jumped up onto the trailer, sat down, and winked at me. He and the security guard had great fun with the trailer, asking me jokingly if I needed help parking. In their opinion, they had to keep me at the gate for a while since they doubted my ability to park the trailer on my own. After a while I understood from their tone that this was a way of flirting with me. First, I was rather offended. I had told them about my research. Then I decided to play along. I smiled and told the police officer he was probably right, but that I thought he should give me a chance to park. The police officer and the security guard had a conversation while they looked at me. The police officer told me to wait and then uncoupled the trailer from the car, sat down inside it and ate his lunch. The security guard was circling around and then he too jumped up onto the trailer. Finally, they decided to give me a chance to try my hand at parking and let me go. I saw the security guard talking in his walkie-talkie and then waved me through.

The story with the trailer does not end here, but the interaction between the involved parties needs to be reflected upon. First, I interpret the interaction between the two men as a contest of displaying hegemonic masculinity [18]. They literally show their "muscles" to me, the female researcher, and they also use their power as both security guard and police officer to hold me back. Secondly, all those activities are done with a great deal of humor and charm. They play out their part of their erotic capital using social skills, liveliness, and social presentation in this flirtation scenario [8]. My response—by laughing and playing along—allows them to go on and not lose their face. I could have stopped the scene by, for instance, becoming angry, but I chose not to because I was interested in observing the

interplay between the two men. At the same time, I felt powerless. They took my trailer and they were also the gatekeepers, so my entry to the festival was in their hands.

When I was finally able to drive into the festival area, I was stopped by two male parking stewards. They wanted to help me park the car. I told them I could manage this myself. Just as I was about to park, the trailer was suddenly gone. In my rear-view mirror I saw the two young stewards running away with it. It looked like a competition between them, both trying to run as fast as possible, clinging on to the trailer. When they were finally exhausted, they brought it back in the parking space and sat on it. It became clear to me that the trailer was functioning as a symbol of manliness, and as an opportunity to play out a certain type of masculinity.

By bringing this symbol with me as a female researcher, it clearly triggered some young men to show their "masculine selves." I became the audience for this male competition, and I was expected to applaud their masculine performances. The story of the trailer also shows how a masculine symbol is put into contest. When leaving the festival, I had to bring the trailer with me. A group of young boys offered their assistance both in carrying and packing my luggage. I was simply supposed to sit and wait for them to fix everything. Again, I was put in the role of a helpless or vulnerable woman who needed men to fix things for her.

The story of the trailer made me reflect upon how this incident would have affected me if I was an ordinary female participant rather than a researcher. At all of the festivals, it was easy for me to get in touch with the abovementioned type of young men with a clear flirting agenda, but I also tried to get in touch with more female participants to get a better understanding of how they were adapting to and experienced the flirting game.

### 3.2. Erotic Capital and the Body

Erotic capital is connected to how one displays one's body. During my fieldwork, I noticed that the female body was being rated by men. On the festival stage, there also seemed to be stricter frames for female behavior than for male. The female performances on stage were closely linked to their looks. One example is how one of the few front women was presented. Every time she entered the stage, the band plays the first line of the well-known song "Isn't She Lovely" by Stevie Wonder. Only the female leaders were presented in this way. One of the male front men was presented as "the boss of all bosses." The presentation of the females was linked to their body and their looks. Most of the women on stage were dancing and singing, while the men were managing the meetings and preaching. Goffman describes this way of displaying male and female power as "relative size" [23]. The presentation of a male leader as "the boss of all bosses" gave him a very different status and position than the female leader, who was presented as "lovely."

As the sociologist Linda Woodhead put it: "Women may be allowed to enter public space as never before, but the pressure on them to look attractive to men when they do so continues to prevent them doing so on their own terms" [24] (p. 152). The female participant has to be *seen.* To do so they have to present themselves as "erotic symbols" by using make-up, feminine clothing, showing the best parts of their body, and so on. At the same time, they have to behave in a way that makes them available to the boys, while still being appropriate. They have to attract the desired attention, but not cross the line. The female body is subject to a double standard. In one way, they are supposed to act decently, while, in the other, act out their erotic capital in order to be considered attractive in the flirting game. The female body is balancing between the extremes of "whore" and "Madonna."

The male body is disciplined in the same way as the female. It seems to be "unavoidable" for men to express their lust and sexual needs. This can be connected to a view of masculinity as inherited, delineated by unchangeable characteristics:

"True masculinity is almost always thought to proceed from men's bodies—to be inherent in a male body or to express something about a male body. Either the body drives and directs action (e.g., men are naturally more aggressive than women; rape results from uncontrollable lust or an innate

urge to violence), or the body sets limits to action (e.g., men naturally do not take care of infants; homosexuality is unnatural and therefore confined to a perverse minority)" [18] (p. 44).

The flirting game is also a heterosexual game. At the festivals, sexuality is presented as a legal act within the institution of marriage between man and woman. Outside of marriage, one has to control the body so that lines are not crossed. Gender and sexual orientation function as disciplining factors that sort normal bodies from abnormal, sinful bodies from pure.

### 3.3. Strategies for Adapting to the Flirting Arena

During my fieldwork at the different festivals, I realized that the participants had different strategies in adapting to the flirting arena. At the same time, they also had to adjust to the festivals' religious views regarding, for instance, sexuality. Not everyone does, and some of the festivals have security guards to make sure that boys and girls do not sleep in the same tents. Some always manage to sneak out and break the rules. Every year a few young participants get expelled from the festivals for breaking of the "tent rules," for drinking alcohol or other kinds of violations. To be able to participate in the flirting game, the participants have to play their cards right. They need to behave in a "suitable" way, so they do not get expelled from the festival arena. At the same time, many of the young boys and girls tell me that one of the reasons that they travel to these festivals, in addition to the religious focus of the festivals, is to find themselves a boyfriend or girlfriend. While talking to different young girls and boys, I understood that the way the flirting game was constructed at these festivals created both "winners" and "losers." Further, the same kinds of flirting strategies seemed to be found across the different festivals. Other studies of flirting and sexuality point to the same. There is a certain male and female mode that is being repeated: the boys' focus on the thrill and the idea of a romantic relationship is just one of *many* different forms of expressing male sexuality. For women, the romantic relationship and intimacy is the (almost) only way of framing actions of sexuality [1,25]. In the following, I will outline three different ways the boys and the girls tried interacting in the flirting game. These are "typical" strategies that I observed during my fieldwork. I am aware that, by bringing in these strategies, I am not embracing all the different ways flirtation can take place, but I will argue that the strategies sum up similar interactions between boys and girls at all of the festivals.

The suitable

*Interviewee*: My friends and I aren't the kind of girls who flirt with guys: we get flirted with.

*Researcher*: What do you do to get flirted with, then?

*Interviewee*: Well, we walk around smiling and stuff.

A lot of the girls I met at the festivals are concerned with the importance of not being seen as the one to "start the flirtation." They get flirted with. They are innocent, kind, and rather shy, and it is the guys who address them. In the flirting game, their role is in the first instance to be the passive ones, and to be "discovered." In order to be discovered, it is important to look good, and that can be hugely challenging when you are on a camping trip. The ladies' room is always full. However, the men's room is rarely crowded. The girls go to a lot of trouble. They have to put on make-up, but not too much; dress cool, but not too cool; look, but not stare. It is not so easy to attract the attention you desire. At some festivals, girls are in the majority and some of them work extra hard to gain attention:

The preaching session has started. The preacher is on stage talking. Suddenly there is a lot of noise at the back of the meeting hall. A gang of teenage girls comes giggling through the hall. They are wearing a lot of make-up, mini-skirts, and crop-tops. They stop in the middle of the room. Soon they have all the boys' attention. They look around. A group of boys wave and whistle at them and signal for them to come and join them. The girls look at each other. They accept the boys' offer. One of the girls sits on a boy's lap. Another one shares a chair with a boy. By the time the worship songs start, all of the girls are making out with the boys (field note).

These girls have spent a long time getting themselves ready. They clearly want to be seen, and they manage this by showing up too late for the meeting and making their entrance with clothes, make-up, and looks in a way that attracts the attention of the young boys in the hall. After that, it is up to the boys

to make their move, to open the flirtation game. The girls take the boys' offer into consideration, then sit down and allow the boys to touch and kiss them. These girls openly demonstrate their availability to the boys. At the same time, they follow quite traditional gender patterns. They wait for the boys to make the first move. The flirtation game has a gendered "division of work" where the boys are expected to take the initiative [1,26].

The partly shy, beautiful girls get a lot of attention. They are pushed, nudged, and flirted with, and they know the codes, they play ball, and know very well how to react. For the really shy girls, things can be more difficult. One girl says:

Me and my friends have always been, you know, very shy. We weren't the kind of people who would easily get in touch with guys. Then there was the cool-girl group, who went together as a group with the cool guys that we also liked. They were like, I don't know, I would feel really cool if one of them would talk to me."

The same girl goes on to tell how she did not understand the "codes":

I started pretty early as a leader. I was about 16–17, so I was kind of the same age as the other participants. There were a lot of male leaders that were a lot older than me. You would look up to them, the ones that were older. At some point I became aware that I've always loved dressing up a bit—or rather to wear mini-skirts. Without ever making much of it. I just liked it. And I did this at the camps as well. And then I got some comments from the male leaders. Like, the kind of on-the-edge comments, which no one would really say.

*Researcher*: How did you react to that?

Well, I remember getting this, kind of weird feeling. Kind of ... But I didn't stop wearing mini-skirts because of that. I found that out myself later. I was probably very much outside of those things. I didn't understand how men reacted to such things. How they looked at me. I never noticed. But I've understood it now [laughs].

At this festival, the mini-skirt became a symbol for the male leaders, which they used to sexually harass the female leader. The mini-skirt can be interpreted as a feminine symbol, quite in contrast to i.e., the trailer, as mentioned before. In both cases, the male leaders used their hegemonic position to suppress women. The Canadian sociologist Sonya Sharma did a study of young Christian women and their views on sexuality. One of her findings was that the women "experienced shame and guilt when they did not experience themselves as being a 'good Christian girl,' someone who follows the 'rules,' who remains sexually 'abstinent'" [27] (p. 75). In the case of the mini-skirt, it is the male leaders who characterize the young female leader due to her clothing. Their harassment strategies can also be seen as a way of disciplining sexuality and "( ... ) especially women's sexuality, keeping it under patriarchal control" [27] (p. 75).

At the festivals, girls are supposed to be rather innocent and shy, but not too much. They have to know what is going on, to join in the "game" with the active boys. But what happens when both are shy? How are the shy boys looked upon?

A girl participant says:

I was so in love with Per. We went together to the same group, and I had been in love with him so for long. I didn't dare say anything to him, I was very shy and afraid to ruin our friendship. So, I learned about this festival, I really wanted to go. I mentioned it to the rest of the gang, but no one really wanted to go, except Per. It was fantastic. We went to the festival together. I was so in love that I nearly died. We were going to stay together at the festival as well. It was kind of the cheapest alternative. I didn't get much out of the meetings, I just sat there and thought about him all the time. I hoped he was going to say something to me, touch me, anything. But nothing happened. My thoughts were so consumed with thinking about him that I went on to praying to God that something would happen several times. I can't understand how he couldn't realize that I was in love with him, but he didn't mention it, and I couldn't say anything. It was so hard ...

This girl feels that it is the boy's responsibility to take the initiative. The boy does nothing and therefore nothing happens. The initiative to opening the flirting game is up to him. Some of the female

participants refers to shy boys as "sissies" and as not very interesting; if guys will not dare to get closer, even while the girls' strategies have opened the door for it, they are regarded as not being potential contestants in the flirting arena. The shy and passive boys enter into a role that is reserved for the girls and become, in a sense, the losers in this ritual game.

The spiritual

"At the gender-market some people climb all the way to the top in success; some fall down; and some never even come close" [26] (p. 98). Those who never get evaluated at the flirting arena do have other opportunities at the festivals. They can "resign" and instead seek a spiritual career. Even though they may not get the attention they want from the opposite sex, some might choose this direction—there is, after all, the idea of a loving god, one who loves them no matter what and wants to be near them. There can, of course, be many different reasons why someone would resign from the flirting game. However, a lot of those I would call spiritual still take part in flirting, but in a more covert way. The flirting is dressed up in many different euphemisms: it becomes spiritual. It is no longer about flirting; it is about God's intervention. It is God who will let you meet that person, and he will give clear signs for it. One of the boys tells the story of how he met his girlfriend:

I fought a lot with myself there. I was very minded that I wasn't going to get together with the wrong person. And I can remember this one time, I actually prayed that if I was meant to be with her, than she would have to call me or send me an SMS before a certain time, or the time I had prayed about, and it was very unlikely she was going to do so. It was only a few minutes notice (to God). So, I kind of thought it to be rather special, and in the exact moment the clock turned to that time, a message popped up on my cell phone. And I had two of these things happen to me. And the other one was, if she would pick something up, it was really funny, but it was a thing that couldn't really happen anyway, but it did, and I interpreted that as a sign that she was the one [laughs].

Another girl tells me about a guy she had "something going on with... But it never evolved into anything, because he wanted to be 100 per cent sure that it was God's will... He figured out these crazy ideas, almost as though if the lightning didn't strike it wasn't God's intention," she explains. It never turned into a relationship and she stopped thinking about it. The same mode as in the other flirting strategies is also present here. Even though God's will was involved, it is the boy who gets the message. The girls are supposed to wait until the signal is clear.

The active

The third strategy in flirting game is being the active one. This is almost exclusively a male strategy. The girls can play along or reject the guys. In my research, I met a lot of boys employing this strategy. Some of the boys told me that they only attend Christian festivals in order to flirt with girls. They don't attend any meetings.

The boys at the festivals are quite creative in their attempts to flirt with the girls. Some have watchmen standing duty outside their tents to bring in girls; some have giant speakers and placards urging girls to visit their tents. Outside one of the tents some of the boys had set up a "tripwire" across the road. Every time a potential girl walked past the guys would pull the rope, and they would fall over. This way they made contact with many girls.

Some of the boys I have talked to also refer to girls who are active participants in opening the flirting game. Some girls break with the standard "rules" for flirting. "Last year there was this girl who was in all the boy-tents," one of the boys tells me. He and his friends are not attracted to girls like that. "It's just wrong if you're only here to date." If the girls become too active, many of the boys will back off. The girls are referred to as "not good" and their behavior is considered to be wrong. These girls may also be referred to as wrongly dressed, "cheap" and "not good Christians." It is interesting that, in some cases, the mini-skirt or crop-top t-shirt can represent erotic capital, but, in other cases, are considered "cheap" or even "slutty." How the clothes are interpreted as erotic capital deeply depends upon how the girls manage to follow the gendered rules of the flirting game.

## 4. The Trailer, the Flirting, and the Gendered Game

In this article, I started with the stretching out and trailer stories to show how I as a researcher was introduced to the flirting game. I have further tried to elaborate various strategies in the flirting that indicate that this game is activity-characterized by strict and traditional gender roles. In order to understand the flirting strategies, it can be fruitful to turn to Goffman's view on gender as a social framework. Goffman views the frames as socially designed and therefore changeable [7]. People can choose to break the general expectations associated with a role; one can consciously or unconsciously lose face.

In the flirting game, the boys are supposed to be the active part [28] (p. 58). The winners are the ones who manage the rules and who play their cards right. The cards of the game are represented by different types of gendered behavior and symbols. The girls have to make an investment in their body and their looks in order to be considered by the boys. It is the boys who have the first choice [28] (p. 58). The girls are rated by their looks and behavior. In some cases, this is also true for the boys, but this happens after the boys have made the first move and initiated the game. The girls, in turn, can then reject or embrace the boys' approaches.

At the Christian festivals, flirting takes place within a heterosexual framework. To be part of this game, the girls are supposed to be nice, feminine, and available. The boy's role is to be active and take the initiative in the flirting game. If the girls become too active or the boys too shy and passive, they will both be ignored by the opposite sex. Boys and girls are exposed to constant evaluation; the worst outcome, however, is not to be evaluated by anyone at all. Both sexes work hard to become a participant in this game of attention-winning, representing two different worlds for boys and girls. There are both male and female losers. Certain roles are overlooked and placed on the sideline. This happens both for the shy and passive boys and also for the girls who become too active. Nevertheless, the gendered rules seem doubly strict for the girls. They have to manage both their own and the boys' lust. It is therefore interesting why there is a majority of female participants in these Christian festivals. The focus on romantic love, family values, security, and marriage have been pointed out as some possible explanations as to how women outnumber men in Christian environments, even though men still have the formal power positions [24,29–31]. By bringing the girls into passive positions, they are also under male dominance. The female participants' erotic capital is connected to looks, attractiveness, and symbols of self-presentation. For the male participant, it is more about taking the lead in the flirting game. As previously discussed, the notion of erotic capital is closely connected to power relationships. That brings me back to the trailer. Female domination over masculine symbols is not accepted in this context. The case of the trailer also shows that there is still a certain type of male behavior—hegemonic masculinity—that seems to gain the most benefit from the flirtation game. The study of Christian festivals shows that there is a need for more in-depth research that focuses on views on sexuality in Christian environments, as well as the role of the researcher in the "flirtation field."

**Funding:** This research received no external funding.

**Conflicts of Interest:** The author declares no conflict of interest.

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
