# Peer review of "The Trailer as Erotic Capital. Gendered Performances—Research and Participant Roles during Festival Fieldwork"

_societies, doi:10.3390/soc9040083_

Round 1
Reviewer 1 Report
Thank you for giving me the chance to read this inspiring paper. It is well-written and thoughtful. The theoretical framework of erotic capital and hegemonic masculinity works well and offers a novel approach to youth activities in religious context. The author draws effectively on key theorist of performance, Erving Goffman, and on theory of erotic capital by Hakim and hegemonic masculinity by Connell.
I think the topic is important and interesting. There is a relatively small body of research that is concerned with religion and sexuality, especially related to secular Nordic countries. Moreover, focus on flirtation as a social process fills a gap in both youth and religious studies. Here the role of researcher is also reflected, not just as someone conducting the research or as a cognitive actor, but also as an embodied person and as a body present at the festival. This is an asset. Another contribution of this article is that it focuses on temporary context, on festivals, which are very timely phenomena of religiosity.
There are certain points that I would like the author to reconsider and improve.
Outlined focus
The title of the article, as well as the abstract, imply that the focus is on the researcher’s role, and that the article reflects the field relations of him/her. Yet, as I read on, the focus is very much divided: a major part of the article examines the overall role of flirting as a game and as gendered performances between girls and boys (like on pages 7-9). So I advice the author to do some revision to the title and the abstract so that they reflect better the content of the paper.
Introduction
In this chapter, it would be useful to have a very short outline of what are these festivals and who are these young people. Although these are clarified later more thoroughly. The sentences describing the empirical material and the events on page 4, in the first paragraph of the ”Material and Methods”, could be moved here.
There is some very interesting and illustrative data in this article, but the author should pay more attention to the niche already in Introduction so that the readers could grasp the significance of the findings. Why is it important to study these flirtation strategies appearing among youth at these festivals?
Related to that issue above: the outline of the concept of flirtation would benefit from connecting it to parallel concepts. Why are these encounters defined as flirtation? Why not sexual harassment, or bullying (this came to my mind when I read the trailer incidence)? The relationship (or not) with the sexual harassment would deserve a comment also because the Me-too-campaign has been visible in the Nordic countries (author discusses the sexual harassment partly on page 8 which supports mentioning it here).
A microsocial approach on the study of flirting…
One who has widely discussed both gender and performances in a ”Goffmanian way” is Judith Butler. She would be a good and necessary complement to the first paragraph: at least a mention of her. She has contributed a great deal to the feminist approach.
Materials and Methods
Author should add some sentences describing the data and handling of it: observation, interviews, transliteration, analysis?
As a strength of the study, observation was conducted at several, different festivals, some were attended even twice by the researcher. Consequently, analysis is plausible. The author lists the festivals but, as I can see, the fourth festival is not listed. The text goes from ”the third” to ”the fifth”. This should be revised.
Results
If the author would like to contribute more to the studies of religion, I think that more could be made of the religion’s impact (or not) on the space and possibilities to flirtation, as well as on the patriarchal hegemony in this context. For instance, one could make an interpretation that perhaps the religious context gives certain security, so that the researcher, close to the age of young participants, experiences the boys' suggestions at 2am just as flirtation? Furthermore, how much the religious views affect, for instance, when ”there also seemed to be stricter frames for female behavior than for male” (page 6)? The author claims once on page 7 that:” At the same time, they also had to adjust to the festivals' religious views.” How? I think, all this merits some consideration in ending chapter.
Based on Hakim’s statement ”… how these gendered relationships can be also empowering for a female position” (line 112), I could argue that these cited conclusions below, on pages 8-9, tend to point in that direction: ”They want to be seen” (girls), ”, ”… they know very well how to react”, ”Some of the female participants refer to shy boys as ”sissies”…”. So my question is: are the boys really one-sidedly on top of the flirting game, initiatives or active and girls passive, as you state in conclusions? And further: could the boys also follow the male leaders as role models? In my opinion, these points could be reflected too but I will leave this to the author to consider.
Author Response
Notes to reviewer 1
Thank you very much for a thoroughly reading and constructive comments.
I have paid attention to these thoughtful comments and tried to incorporate them in the article as follows:
Outline focus: I have changed the title and brought the participant focus into title and abstract.Introduction:
Paragraph for “materials and methods” are moved to this part.
Line 57-72 Trying to outline to need for this study and a deeper focus on the religious context
The Butler perspective and also some of the critique of Goffman´s gender perspective is brought to attention in line 115-125
Line 217-219 Bringing in the concept of sexual harassment.
Methods:
Line 237-297 Describing the data and handling of it.
Line 302 elaborated the third festival
Results:
The festivals religious views: line 446 – 458 trying to outline the relation between the religious and flirting focus
Line 412-413 Trying to elaborate the female roles on stage
I have tried to elaborate that there also are male losers in the flirting game. Especially the shy and passive boys line 642-643
Reviewer 2 Report
Thank you for the opportunity to read the manuscript.
Some comments/suggestions:
Present a title more clear to the text context; Keywords are not clear. It seems necessary a reformulation; Pertinent theoretical framework. Congratulation (its fair); For an international reader, the explanation in some details the specific characteristics of these festivals (Ln 133-...) is essential and sufficient; Explain the methods and the analysis of data process; the references need to be improve (only 26); In my opinion, manuscriptgives the idea that is a regularity this erotic dimension.It's usual and general to the young people that participates in these festivals? What the importance of to be "Christian festivals"?
Author Response
Notes to reviewer 2
Thank you very much for a thoroughly reading and constructive comments.
I have paid attention to these thoughtful comments and tried to incorporate them in the article as follows:
Line 48-70 Trying to outline to need for this study and a deeper focus on the specific religious context of these festivals and also the focus on flirting and also in line 445-448
I have focused on improving comments on methods and design.
Line 237-297 Describing the data and handling of it.
About the keywords: I will leave it up to the editors to decide, but I would like to keep the chosen keywords.
Reviewer 3 Report
Although the subject proposed by the author is interesting, I think that it is necessary to provide as much objective information as possible to reinforce their hypothesis. In fact, I can only see the biased opinion of the authors instead of objective data that reinforce their hypothesis.
Author Response
Thank you so much for taking the time to read my article and respond to it. I have focused on improving comments on methods and design. I have also tried to present the material as objective as possible, also by focusing on my role as a female researcher.
Round 2
Reviewer 2 Report
Now the paper seems to be more rigorous and understable for an international reader. Please, verify the readability.
Author Response
The article has been edited by a native english speaker.